# EEG Patterns in Patients with Prader–Willi Syndrome

**DOI:** 10.3390/brainsci11081045

**Published:** 2021-08-06

**Authors:** Maurizio Elia, Irene Rutigliano, Michele Sacco, Simona F. Madeo, Malgorzata Wasniewska, Alessandra Li Pomi, Giuliana Trifirò, Paolo Di Bella, Silvana De Lucia, Luigi Vetri, Lorenzo Iughetti, Maurizio Delvecchio

**Affiliations:** 1Oasi Research Institute-IRCCS, 94018 Troina, Italy; luigi.vetri@gmail.com; 2Pediatric Unit, Casa Sollievo della Sofferenza, San Giovanni Rotondo, 71013 Foggia, Italy; i.rutigliano@operapadrepio.it (I.R.); m.sacco@operapadrepio.it (M.S.); 3Pediatric Unit, Department of Medical and Surgical Sciences of Mother, Children and Adults, University of Modena and Reggio, 41124 Modena, Italy; simonamadeo@hotmail.com (S.F.M.); lorenzo.iughetti@unimore.it (L.I.); 4Department of Human Pathology of the Adult and of the Developmental Age “Gaetano Barresi”, University of Messina, 98124 Messina, Italy; mwasniewska@unime.it (M.W.); alessandra.lipomi92@gmail.com (A.L.P.); 5Cardiogenetic and Vascular Center, IRCCS Policlinico San Donato Milanese, 20097 Milan, Italy; gtrifi@libero.it; 6Division of Child Neurology and Psychiatry, “G. Martino” Hospital, University of Messina, 98124 Messina, Italy; paolo.dibella@polime.it; 7I-Motion-Pediatric Clinical Trials Department, Trousseau Hospital, 75012 Paris, France; silvidelucia@hotmail.it; 8Metabolic Diseases and Genetics Unit, Giovanni XXIII Children’s Hospital, 70126 Bari, Italy; mdelvecchio75@gmail.com

**Keywords:** EEG, Prader–Willi syndrome, wakefulness, sleep, epilepsy, genetics

## Abstract

Prader–Willi syndrome (PWS) is a rare disease determined by the loss of the paternal copy of the 15q11-q13 region, and it is characterized by hypotonia, hyperphagia, obesity, short stature, hypogonadism, craniofacial dysmorphisms, and cognitive and behavioral disturbances. The aims of this retrospective study were to analyze interictal EEG findings in a group of PWS patients and to correlate them with genetic, clinical, and neuroimaging data. The demographic, clinical, genetic, EEG, and neuroimaging data of seventy-four patients were collected. Associations among the presence of paroxysmal EEG abnormalities, genotype, and clinical and neuroimaging features were investigated. Four patients (5.4%) presented drug-sensitive epilepsy. Interictal paroxysmal EEG abnormalities—focal or multifocal—were present in 25.7% of the cases, and the normalization of the EEG occurred in about 25% of the cases. In 63.2% of the cases, the paroxysmal abnormalities were bilaterally localized over the middle–posterior regions. Brain magnetic resonance imaging (MRI) was performed on 39 patients (abnormal in 59%). No relevant associations were found between paroxysmal EEG abnormalities and all of the other variables considered. Interictal paroxysmal EEG abnormalities—in particular, with a bilateral middle–posterior localization—could represent an important neurological feature of PWS that is not associated with genotype, cognitive or behavioral endophenotypes, MRI anomalies, or prognosis.

## 1. Introduction

Prader–Willi syndrome (PWS) is a rare genetic condition with an estimated prevalence of about 1:15,000 that is characterized by multisystemic features, such as severe early hypotonia, hyperphagia, childhood obesity, short stature, small hands and feet, hypogonadism, growth hormone or other endocrine deficiencies, craniofacial dysmorphisms, developmental delay, intellectual disability (ID), behavioral disturbances, and autism spectrum disorder [1].

PWS is a genomic imprinting disorder, as it occurs due to the loss of a paternal copy of the 15q11-q13 region. Most frequently (in approximately 60% of cases), this is the result of a de novo paternal deletion of the 15q11-q13 region, leading to a lack of expression of paternally derived genes; in about 35% of cases, maternal uniparental disomy (UPD)—with both instances of chromosome 15 inherited from the mother—is present. Imprinting center (IC) defects (micro-deletions, mutations) in the 15q11-q13 region or other chromosomal abnormalities are found in the remaining 5% of PWS patients [2].

Until now, no peculiar interictal electroencephalographic patterns have been identified in PWS, as they have in other chromosome abnormalities, and heterogeneous EEG pictures have been reported, such as focal, multifocal, or generalized epileptiform abnormalities [3,4,5,6,7,8,9].

Although epilepsy does not represent a typical neurological feature, recently, a prevalence ranging from 4% to 26% was estimated in several retrospective PWS series worldwide. Seizures usually start before two years of age, and epilepsy is mostly of the focal type, but generalized epilepsy can also occur, with generalized tonic–clonic seizures and, more rarely, atypical absences and atonic seizures [3,4,5,6,7,8,10,11,12]. Febrile seizures occur in 6.4–39.2% of children with PWS [7,8,12]. A definite association between the prevalence of epilepsy or the seizure type and the genotype in PWS has not been demonstrated; in fact, some studies reported a significantly higher frequency of seizures in the deletion subgroup than in the UPD subgroup [3,5,10], but others did not confirm that finding [6,7].

Specific EEG patterns were previously described in a series of genetic disorders [9], but not in PWS. Therefore, the main hypothesis of this retrospective multicenter study was to define a possible peculiar EEG trait in this genetic condition by analyzing the interictal EEG findings. In addition, we tried to investigate any associations between the EEG findings and some relevant clinical and neuroimaging data, such as data on the genotype, cognitive level, behavioral disturbances, and brain MRI abnormalities.

## 2. Materials and Methods

We retrospectively recruited subjects with PWS referred to 4 Italian Referral Centers, i.e., Oasi Research Center (Troina, Italy), the Pediatric Unit of Casa Sollievo della Sofferenza (San Giovanni Rotondo, Italy), the Pediatric Unit of University of Modena e Reggio (Modena, Italy), and the Department of Human Pathology of the Adult and of the Developmental Age “Gaetano Barresi” (University of Messina, Messina, Italy).

All of our patients were representative of all PWS patients because they were initially diagnosed and then clinically managed at centers that specialized in pediatric endocrinology and clinical genetics. The necessary inclusion criteria of this study were an established genetic alteration that was typical of PWS and the availability of EEG recordings. In the above-mentioned centers, the EEG was a common diagnostic tool included in the neuropediatric workup of developmental disorders and rare comorbid diseases, such as PWS, even in the absence of seizures.

This study was approved by all of the local ethics committees of the centers in which the study was performed. Written informed consent was obtained from the patients’ parents or from the patients, as appropriate. All investigations were conducted according to the principles expressed in the Declaration of Helsinki.

The clinical diagnosis of PWS was made according to the consensus clinical diagnostic criteria [13] and was confirmed by genetic testing, by means of methylation analysis and FISH, or by means of analysis of the parental inheritance pattern of chromosome 15, when indicated, according to the recommendations of the American Society of Human Genetics/American College of Medical Genetics [14].

For all of the subjects, we evaluated their gender, age (range and mean) at the first and last EEG, duration (range and mean) of the EEG follow-up, BMI (range and mean) at the first and last EEG, genotype, degree of intellectual disability, behavioral disturbances, presence of seizures, age (range and mean) at seizure onset, seizure type, EEG and MRI findings, and drug therapy. MRI was carried out in order to detect possible pituitary or brain anomalies. When clinically suspected, the presence of sleep apnea was assessed by using polysomnography.

Intellectual disability (ID) and borderline intellectual functioning (BIF) were classified according to the DSM 5 criteria [15]. Seizures were classified according to the measures from the International League Against Epilepsy [16]. Interictal wakefulness or sleep EEG was recorded in all of the subjects according to the International 10–20 system, and all recordings were visually interpreted by an expert neurologist (M.E.).

We investigated the possible correlations between the presence of paroxysmal EEG abnormalities and other clinical and neuroimaging features, such as the degree of cognitive dysfunction (subjects with BIF or mild ID versus subjects with moderate or severe ID), behavioral disturbances (presence versus absence), genotype (deletion 15q11-q13 versus UPD), and brain MRI abnormalities (presence versus absence).

A statistical analysis was performed by using SPSS version 20.0 for Mac OS (IBM Corp., Armonk, NY, USA). The age did not follow a normal distribution (Kolgomorov–Smirnov test), and thus, a non-parametrical Mann–Whitney U test was used for comparisons between groups. The Yates-corrected chi-square test was used to assess any differences between categorical variables. The data are presented as the range and mean ± SD or rate. *p*-values < 0.05 were considered statistically significant.

In addition, we checked for possible simultaneous associations of age, presence of behavioral disorders, pathological findings in MRI, and genetic conditions, which were used as independent factors/predictors; abnormal or normal EEG or the level of intellectual disability was considered as the dependent variable. This was achieved by means of the General Regression Models module offered by the commercially available STATISTICA v.6, StatSoft Inc (Tulsa, OK, USA). This module allows one to build models for designs with categorical predictor variables, as well as with continuous predictor variables. For the dependent variable, the statistical significance of the association of each independent factor was obtained by considering the effects of the other independent factors.

## 3. Results

### 3.1. Patients Features

We studied 74 subjects with PWS who were aged 2–42 years; 39 were males (52.7%) and 35 were females (47.3%). Thirty-two patients (43.2%) had a 15q11-q13 paternal deletion, thirty-eight (51.4%) had a maternal UPD, and four (5.4%) had an imprinting center defect (ICD). ID was diagnosed in 67 subjects. Four patients (5.4%) presented epilepsy with various types of seizures, such as absences, focal seizures with impaired awareness, or generalized tonic–clonic seizures. One patient had febrile convulsions. Seizure onset ranged from 1 year and 3 months to 7 years, and in all cases, seizures were controlled with antiepileptic drugs in monotherapy or polytherapy. Sleep apnea was diagnosed in 17 patients out of the 28 who underwent polysomnography. The following drugs were given to treat seizures and/or behavioral disturbances: topiramate in five cases, valproate in three cases, levetiracetam in two cases, and oxcarbazepine in one case. The complete demographic and clinical findings are reported in Table 1.

A total of 59 subjects (79.7%) took drugs during the follow-up period, i.e., growth hormone or other hormones (49 cases), hypoglycemic medications (6), antiepileptic drugs (11), and psychopharmacologic treatments (11).

### 3.2. EEG Findings

A total of 198 EEGs were recorded (range of recordings: 1–10, mean: 2.9 ± 1.9, median: 2): 92 during wakefulness and 106 during sleep (Table 2). The age range at the first EEG was 0.08–33 years, with a mean age of 7.7 ± 7.8 years; and the age range at the last EEG was 0.133 years, with a mean age of 10.6 ± 7.8 years. The range of durations of the follow-up was 0.1–19 years, with a mean duration of 2.9 ± 3.9 years. The BMI range at the first EEG was 9.3–67.4 kg/m^2^, with a mean of 24.2 ± 12.1 kg/m^2^; and the BMI range at the last EEG was 13.5–67.4 kg/m^2^, with a mean of 25.6 ± 11.1 kg/m^2^.

In all cases, the background activity was normal for the age. EEG abnormalities were found in 19 patients (25.7%, age range: 0.1–29.3 years, mean: 8.4 ± 8.8), and they were classified as multifocal and middle–anterior (five subjects) (Figure 1) when localized over the frontal–central, frontal–temporal, or central–temporal regions, and they were classified as multifocal and middle–posterior (12 subjects) (Figure 2) when recorded over the temporal–parietal, temporal–occipital, or parietal–occipital regions. Two of these nineteen subjects with EEG abnormalities took psychopharmacological treatments: risperidone and risperidone in combination with lorazepam.

In two cases, focal slow waves were present. In five out the nineteen patients (26%), the EEG picture became normal at the end of the follow-up at 9.01 (range: 4.7–16.3) years of age and 3.6 (range 1–11.3) years after the first abnormal EEG pattern (Figure 3). Only one subject with EEG normalization took antiepileptic medication (valproic acid). The age in the fourteen patients with persistent abnormal EEG patterns was 0.1–29.3 years, with 9.8 ± 9.4 years at the first recording and an age range of 0.1–36.6 years with 11.1 ± 10.6 years after the last one.

### 3.3. Brain MRI Findings

Brain MRI was performed in 39 subjects (Table 3). In 23 of them (59%), brain abnormalities were present: myelination anomalies (4), corpus callosum hypoplasia (3), pituitary hypoplasia (9), subarachnoid space enlargement (7), enlargement of the lateral ventricles (9), arachnoid cyst (5), cerebral atrophy (5), or cerebellar hypoplasia (2). Only two patients with seizures received a brain MRI; one had myelination absence anomalies and subarachnoid space enlargement, and the other had absences that showed pituitary hypoplasia, too. The brain MRI anomalies did not differ in the subgroups with deletions of the 15q11-q13 region and UPD.

### 3.4. Phenotype and Genotype Characteristics of PWS Patients with EEG Paroxysmal Abnormalities Versus PWS Patients without EEG Paroxysmal Abnormalities

The mean ages were, respectively, 11.2 years (range: 3–33 years) and 10.3 years (range: 0.1–33 years) in the subgroups of PWS subjects with paroxysmal EEG abnormalities (PWS EEG+; *n* = 19) and without paroxysmal EEG abnormalities (PWS EEG-; *n* = 55); behavioral disturbances were present in five PWS EEG+ patients and in 12 PWS EEG- patients. The genotypes were deletion 15q11-q13 in 10 PWS EEG+ patients and 22 in PWS EEG- patients, and UPD in nine PWS EEG+ patients and 29 in PWS EEG- patients; ICDs were evident only in four PWS EEG- patients. Brain abnormalities were found in 6 out of 12 PWS EEG+ patients who performed MRI (2 = myelination anomalies, 2 = pituitary hypoplasia, 2 = subarachnoid space enlargement, 3 = enlargement of the lateral ventricles, 3 = cerebral atrophy); they were evident in 17 out of 27 PWS EEG- patients (2 = myelination anomalies, 3 = corpus callosum hypoplasia, 7 = pituitary hypoplasia, 5 = subarachnoid space enlargement, 6 = enlargement of the lateral ventricles, 5 = arachnoid cyst, 2 = cerebral atrophy, 2 = cerebellar hypoplasia). The main characteristics of the PWS patients with paroxysmal EEG abnormalities and PWS patients without paroxysmal EEG abnormalities are summarized in Table 4.

In summary, no differences were found between patients with and without paroxysmal EEG abnormalities in terms of age (*p* = 0.86), behavioral disturbances (*p* = 0.93), genotype (*p* = 0.66), or brain MRI abnormalities (*p* = 0.77). Finally, as a further confirmation of the results reported above, none of the factors considered (age, presence of behavioral disorders, pathological findings with MRI, and genetic conditions) as independent predictors were significantly associated with abnormal EEG or with the level of intellectual disability as a result of the analysis by means of the General Regression Models module.

## 4. Discussion

Chromosome abnormalities are often associated with epilepsy. In particular, Singh et al. [17] reported 400 different chromosome aberrations that were associated with epileptic seizures and/or EEG abnormalities. However, only a few chromosome anomalies have a characteristic electroclinical pattern, such as 1 p36, 4p16, 6q terminal, or 15q13.3 deletions, trisomy 12p, Angelman syndrome (AS), inv dup 15, ring 20, Down syndrome, Xp11.22–11.23 duplication, and XYY [9].

Wang et al. (2005) carried out an EEG study in 26 out of 50 patients with PWS, and 10 of them (38.5%) showed abnormalities. In particular, five subjects (with or without seizures) presented high-voltage EEG activities at 4–6 Hz, four (with generalized tonic–clonic seizures) showed focal paroxysmal discharges, and one (with atypical absences) showed short generalized discharges of polyspike and wave complexes [3]. In a study by Kamuda et al. (2005), four patients with PWS presented with epilepsy, and focal or multifocal paroxysmal abnormalities were described [4]. In a study of 56 patients with PWS, 10 of them had epilepsy, and 9 were revealed to have EEG abnormalities; in 2 cases, these were specified as focal left parietal discharges and nonspecific spikes [5].

Three subsequent retrospective studies revealed focal, multifocal, or generalized paroxysmal EEG abnormalities, respectively, in 13 of 23 (56.5%), 13 of 22 (59%), and 38 of 38 (100%) patients with PWS and epilepsy [6,7,8]. Furthermore, focal epileptiform abnormalities were found in 12 out of 94 subjects (12.8%) collected in an observational cohort study, and a subclinical electrographic seizure pattern was found in 5 out of 12 of these cases [12].

Multifocal or focal EEG abnormalities were found in 19 (25.7%) of our 74 patients, and epilepsy was found in 4 (5.4%). In only two cases, psychopharmacological treatment with risperidone could facilitate the appearance of interictal EEG abnormalities. These prevalence rates are at the lower end of the prevalence ranges reported in the previous literature: 12.8–100% for EEG abnormalities and 4–26% for epilepsy; however, this apparent discrepancy in the prevalence rate could be due to the different settings, as all of our patients were recruited in centers that specialized in pediatric and endocrinological management of PWS [3,4,5,6,7,8,10,11,12]. In approximately one-quarter of our PWS patients, the EEG picture was normalized, a finding that could be not compared with the literature because, to the best of our knowledge, no EEG follow-up studies have been conducted for PWS until now. The disappearance of EEG abnormalities in PWS seems to be an age-related phenomenon, rather than a drug-dependent phenomenon, similarly to what occurs in other neurodevelopmental genetic syndromes, such as fragile-X syndrome. The EEG normalization in fragile-X syndrome appears to resemble that of benign focal epilepsy in childhood, an age-related epilepsy that is present in the general population [9].

We did not find statistically significant associations between the presence of interictal paroxysmal EEG abnormalities and any other clinical, genetic, or neuroimaging features of the PWS phenotype. However, the interictal EEG abnormalities—and, in particular, their middle–posterior localization—could represent an important and supportive neurological feature of PWS, even though they do not contribute by providing information about the genotype, cognitive or behavioral endophenotypes, possible structural anomalies in MRI, or prognosis.

Although our study seems to confirm that a peculiar interictal EEG pattern does not exist in PWS, the paroxysmal abnormalities are mostly focal or multifocal, and in approximately two-thirds of our patients, spikes were localized over the middle–posterior regions. Thus, this localization could represent a rather common EEG trait that could be considered as a potential marker of PWS.

Both epilepsy and interictal EEG abnormalities are more frequently found in PWS patients than in the general population (3.54%) [18], but they are much less frequent than in AS (85%), which affects the same chromosomal 15q11-q13 region [19]. There is not yet convincing evidence for this discordance, but it can be hypothesized that, in PWS, although the missing paternal 15q11-q13 region includes the genes for the GABA receptor subunit cluster (GABRB3, GABRA5, and GABRG3) receptor, the maternal UBE3A is active, which could explain the lower rate of epilepsy in PWS [12]. Another alternative genetic mechanism for the pathogenesis of EEG abnormalities or epilepsy could be represented by the involvement of other contiguous genes in the 15q11-q13 region.

Brain MRI abnormalities were relatively frequent (59%) and heterogeneous in our cohort of patients with PWS. Most frequently, we found enlargements of the lateral ventricles (39.1%) and pituitary hypoplasia (39.1%). Dilation of lateral ventricles, a non-specific finding, was previously observed in patients with PWS in up to approximately 45% of cases, and it can represent the consequence of lost or reduced growth of white or gray matter, or both [6,8,11,12]. Pituitary hypoplasia or morphological alterations, and even pituitary autoimmunity, are very common MRI findings in patients with PWS, in whom the hyperphagia, hypogonadism, and deficit of growth hormone suggest a possible central hypothalamic/pituitary dysfunction [20,21,22]. However, no relationship has been reported between pituitary hypoplasia and interictal EEG abnormalities or epilepsy in PWS.

Our study has an important strength. The recruitment of a rather large series of patients with PWS, who were all managed for a long period of time in highly specialized centers, allowed us to establish the prevalence of interictal EEG abnormalities with a particular accuracy considering the great number of EEG recordings carried out. On the other hand, some methodological limitations should be taken into account. In fact, this study was retrospective; all patients were managed in pediatric and endocrinological settings, and the sample under study was limited by the rarity of the syndrome, and was not representative of PWS as a whole, with 43% presenting a deletion of the 15q11-q13 region and 51% presenting a UPD, which had an impact on the correlations. Brain MRI was available for approximately 52% of the sample studied; a machine-/deep-learning approach to predicting the subtypes or outcomes in PWS patients based on the EEG, genotype, and clinical and neuroimaging data was not used because these capabilities were not available.

## 5. Conclusions

In conclusion, we showed that about 25% of PWS patients presented EEG abnormalities, irrespective of their genetic defects, gender, and intellectual disabilities, and that about 5% presented epilepsy. The EEG normalized in about 25% of the patients with EEG abnormalities. Our study showed that paroxysmal abnormalities are mostly focal or multifocal and are localized over the middle–posterior regions in two-thirds of the patients. This localization might be considered a rather common EEG trait and a potential marker of PWS in about 25% of patients during follow-up. A peculiar interictal EEG pattern does not exist in PWS. Further large prospective studies are needed to clarify the pathogenesis of electroclinical findings in PWS, and they should also address more strict correlations with molecular genetics and high-definition neuroimaging.

## Figures and Tables

**Figure 1 brainsci-11-01045-f001:**
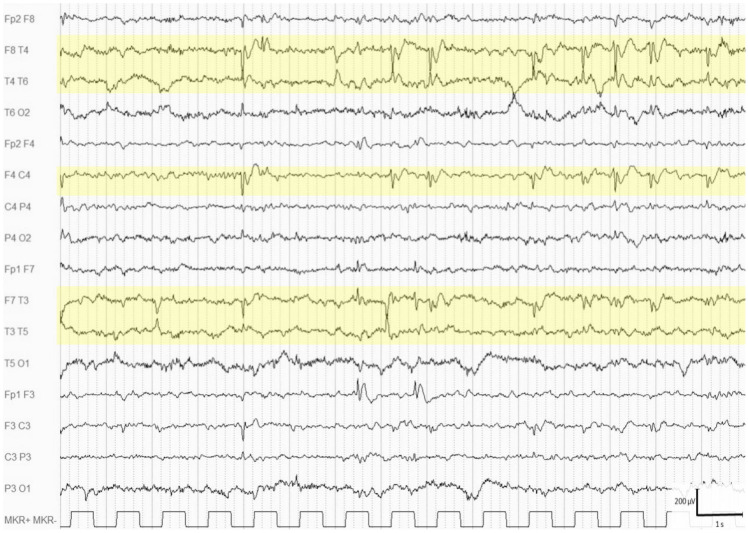
Wakefulness EEG of a 4-year-old boy showing numerous multifocal spikes, more prominent over the right central-temporal and the left frontal-temporal regions (highlighted areas).

**Figure 2 brainsci-11-01045-f002:**
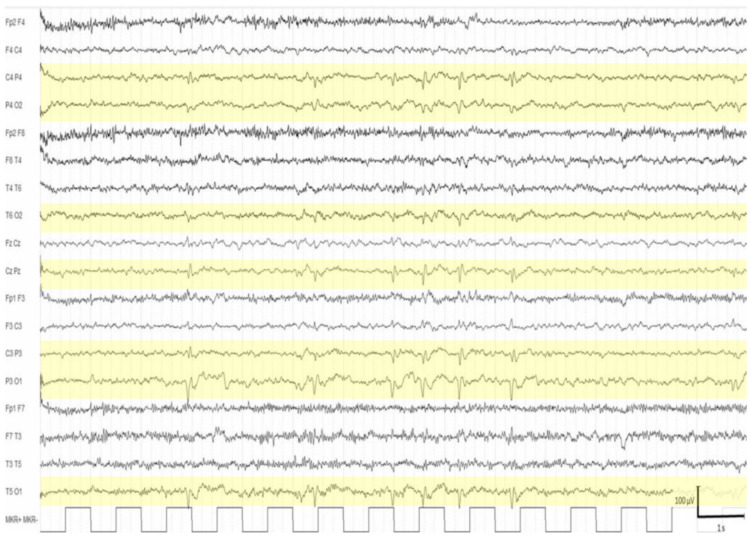
Wakefulness EEG of a 5 year-old boy with spikes synchronous over the parietal-temporal-occipital regions (highlighted areas), prevalent in the left hemisphere.

**Figure 3 brainsci-11-01045-f003:**
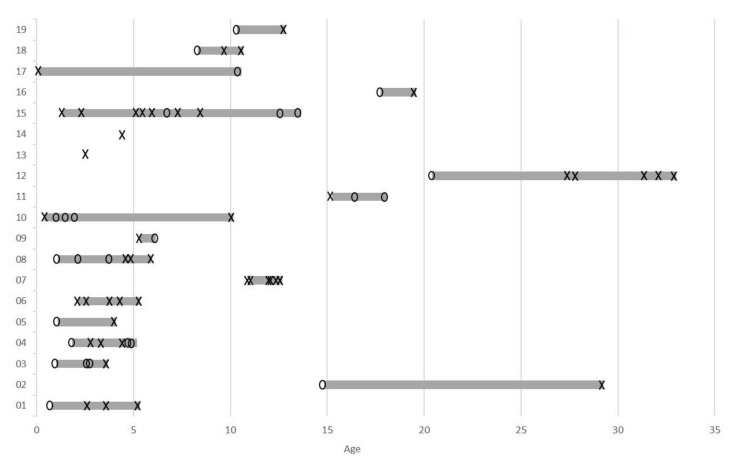
EEG recordings in the 19 patients with PWS and EEG paroxysmal abnormalities. 0 = normal EEG; X = abnormal EEG.

**Table 1 brainsci-11-01045-t001:** Demographic and clinical features of the patients with Prader–Willi syndrome (*n* = 74).

Variables	Results
Gender (males/female)	37/35 (52.7%/47.3%)
Age range (years)	2–42
Follow-up duration (years)	2.9 ± 3.9 (0.1–19.0)
Genotype	
15q11-q13 paternal deletion Maternal UPD ICD	32 (43.2%) 38 (51.4%) 4 (5.4%)
Intellectual disability Mild Moderate Severe	67 (90.5%) 47 (63.5%) 15 (20.3%) 5 (6.7%)
BIF	3 (4.1%)
Behavioral disturbances	7 (9.5%)
Seizures Absences Focal seizures, impaired awareness GTCS	4 (5.4%) 2 1 1
Sleep apnea	17 (23.0%)
Drug therapy	59 (79.7%)

Categorical variables are reported as frequency; quantitative variables are reported as mean ± SD (range). UPD = uniparental disomy; ICD = imprinting defect; BIF = borderline intellectual functioning; GTCS = generalized tonic–clonic seizures.

**Table 2 brainsci-11-01045-t002:** EEG findings.

Variables	Results
Number of recorded EEGs	198 (1–10)
Mean number of recorded EEGs ± SD	2.9 ± 1.9
Age at the first EEG (years)	7.7 ± 7.8 (0.1–33)
Age at the last EEG (years)	10.6 ± 8.3 (0.1–33)
EEG type	
Wakefulness Wakefulness and sleep	92 (46.5%) 106 (53.5%)
Background activity at the last EEG	
Slow Normal	0 (0%) 74 (100%)
Abnormalities Middle-anterior spikes Middle-posterior spikes Focal slow waves	19 (25.7%)
	5 (26.3%)
	12 (63.2%)
	2 (10.5%)

Categorical variables are reported as frequency; quantitative variables are reported as mean ± SD (range).

**Table 3 brainsci-11-01045-t003:** Brain magnetic resonance imaging (MRI) findings in the patients with Prader–Willi syndrome (*n* = 39).

Findings	Number (Rate)
Abnormal brain MRI	23 (59%) *
Myelination anomalies	4 (17.4%)
Corpus callosum hypoplasia	3 (13%)
Pituitary hypoplasia	9 (39.1%)
Subaracnoid space enlargement	7 (30.4%)
Enlargement of the lateral ventricles	9 (39.1%)
Arachnoid cyst	5 (21.7%)
Cerebral atrophy	5 (21.7%)
Cerebellar hypoplasia	2 (8.7%)

* some patients had more than one MRI abnormality.

**Table 4 brainsci-11-01045-t004:** Phenotype and genotype characteristics of PWS patients with or without EEG paroxysmal abnormalities.

	PWS EEG+ Patients	PWS EEG- Patients
Mean age (years)	11.2 (range 3–33)	10.3 (range 0.1–33)
Behavioral disturbances	5	12
Deletion 15q11-q13	10	22
UPD	9	29
ICD	0	4
Brain abnormalities	6	17

## Data Availability

Data available on request due to restrictions e.g., privacy or ethical. The data presented in this study are available on request from the corresponding author. The data are not publicly available due to privacy policy.

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
