# Peer review of "EEG Patterns in Patients with Prader–Willi Syndrome"

_brainsci, 2021, doi:10.3390/brainsci11081045_

Round 1

Reviewer 1 Report

This manuscript from Maurizio Elia and collaborators reports the results from a retrospective stury of EEG patterns in patients with Prader-Wily syndrome (PWS). They investigate 1) whether  there is a particular EEG pattern associated with PWS and 2) if there is any association between EEG findings and relevant clinical/neuroimaging data. They show that EEG interictal abnormalities, i.e. focal and multifocal interictal spikes, are observed in a quarter of the patients, mostly in the midle-posterior region bilaterally. They also show no relevant association between the presence of EEG abnormality and other variables such as behavioral/cognitive phenotype, MRI abnormalities or prognosis.

This is a very interesting study that suggests that the presence of EEG epileptiform activity does not correlate with cognitive/behavioral deficits. It therefore highlights the role of aetiology in these deficits. In relation to this point, can the authors determine, using similar statistical methods (general regression models), if MRI findings or genotype are significant predictors of cognitive/behavioral outcome in this cohort? 

Author Response

Thanks for the positive feedback and for the valuable suggestion. We analyzed by means of the general regression model if MRI findings or genotype were significant predictors of cognitive/behavioral outcome in our cohort, as now specified in the Materials and Methods section (see page 8, last paragraph). We found that both MRI findings and genotype were not significant predictors of cognitive outcome in our cohort, and we changed the statement in the Results section (see page 7, last paragraph).

Reviewer 2 Report

This is an interesting and relevant scientific paper summarizing EEG abnormalities among 74 patients with PWS who were recruited from specialty clinics serving this population.  The background reviews pertinent information about seizures occurring in PWS compared to other genetic syndromes, and mentions specific relevance to ch15 (GABA and UBE3A), but there is insufficient information about the clinical relevance of paroxysmal abnormalities noted on EEG.  Does dysrhythmia in certain leads predict cognitive, mood or behavioral alterations?  What may be the clinical correlation of middle-posterior localization present in 2/3 of the PWS patients in this study?  In the EEG excerpts provided in the paper, please highlight and identify the abnormalities noted.  Although there were no correlations of electrical data with PWS genotype, behavior problems, or MRI, there was a tendency toward normalization with age in 25%.  Is this normalization typical in the general population, or is it due to treatment or other drug effect, as many patients with PWS are treated with anticonvulsants?  On the other hand, are there drugs that create paroxysmal patterns on EEG?  The authors do not comment on the fact that their sample is not representative of PWS as a whole with 42% DEL (usually 60-70%) and 51% UPD (usually 30% except in older mothers).  This difference may have had an impact on correlations, especially among DEL.  Although this study is about EEG abnormalities, the authors could have correlated the MRI results by genetic subtype, as this study reports MRI results on 39 patients.  Please present the data from section 3.4 in a table for easier reading and comparison.  Finally, there is no mention of sleep apnea.  Was there a reason why this was omitted?  In conclusion, thank you for this commendable contribution to the PWS literature.  Please see the attached document for editing suggestions.

Author Response

Thanks for the positive feedback. English language was reviewed by an expert in this field. 

Does dysrhythmia in certain leads predict cognitive, mood or behavioral alterations?

Thanks for the valuable comment. We found that none of the factors considered (age, presence of behavioral disorders, pathological findings at MRI, and genetic condition) as independent predictors resulted to be significantly associated with abnormal EEG nor with the level of intellectual disability at the analysis by means of the General Regression Models module (see Results, page 7, last paragraph).

What may be the clinical correlation of middle-posterior localization present in 2/3 of the PWS patients in this study?

One of the aims of this retrospective multicenter study was to define a possible peculiar EEG trait in this genetic condition, analyzing the interictal EEG findings. We previously described peculiar interictal EEG patterns in Fragile X syndrome, Angelman syndrome, and other rare chromosome abnormalities. Although we could not find correlations between EEG abnormalities and cognitive/behavioral alterations, we think that the frequent middle-posterior localization could represent a potential supportive diagnostic marker in PWS (see Discussion, page 8, 4th and 5th paragraphs)

In the EEG excerpts provided in the paper, please highlight and identify the abnormalities noted.

We highlighted the EEG channels showing the interictal epileptiform activities in the figures.

Although there were no correlations of electrical data with PWS genotype, behavior problems, or MRI, there was a tendency toward normalization with age in 25%. Is this normalization typical in the general population, or is it due to treatment or other drug effect, as many patients with PWS are treated with anticonvulsants?

Thanks for the valuable comment. We think that this EEG normalization could be an age-related phenomenon, similar to what occurs in other neurodevelopmental genetic syndromes, such as fragile-X syndrome. Fragile-X syndrome seems to resemble benign focal epilepsy of childhood, typical of the general population. In addition, only one subject with interictal EEG abnormalities in our cohort took an antiepileptic drug (valproic acid), then it is possible to suppose that EEG normalization could be age-dependent, and not a drug-dependent phenomenon. We added a sentence in the EEG findings section (see page 6, 2nd paragraph), and another in the Discussion section (see page 8, 3rd paragraph).

On the other hand, are there drugs that create paroxysmal patterns on EEG?

Only two patients out of 11 who were treated with psychoactive compounds presented paroxysmal EEG abnormalities. One took risperidone, the other one risperidone and lorazepam. We added these details in the EEG findings section (see page 5, 2nd paragragh), and in the Discussion (see page 8, 3rd paragraph).

The authors do not comment on the fact that their sample is not representative of PWS as a whole with 42% DEL (usually 60-70%) and 51% UPD (usually 30% except in older mothers). This difference may have had an impact on correlations, especially among DEL.

Thanks for the valuable suggestion. We commented this issue as another limitation of our study in the Discussion.

Although this study is about EEG abnormalities, the authors could have correlated the MRI results by genetic subtype, as this study reports MRI results on 39 patients.

We correlated MRI results with genetic subtypes by the General Regression Models module of the statistical software, but we could not find any correlation (see Results, page 7, last paragraph).

Please present the data from section 3.4 in a table for easier reading and comparison.

We added Table 4 presenting data of section 3.4.

Finally, there is no mention of sleep apnea. Was there a reason why this was omitted?

The study was retrospective and we had a small number of patients who underwent polysomnography, based on the clinical suspect of sleep apnea. However, we added details on sleep apnea in the Materials and Methods section (see page 3, 3rd paragraph), and in the Results section (see page 3, 1st paragraph). We found that, out of 28 subjects who underwent polysomnography, 17 had sleep apnea. Among these 17, four had EEG abnormalities; this frequency of EEG abnormalities was not different from that in the remaining patients without apnea (chi-square 0.053, p=0.873). For this reason, we preferred not to comment further on this point.

In conclusion, thank you for this commendable contribution to the PWS literature.

Thanks for the positive feedback and for the valuable editing. We took in count all the suggestions. We added a sentence in the Materials and Methods section (see page 3, 3rd paragraph) to specify that the clinical indication for brain MRI was the detection of eventual pituitary or brain anomalies.

We also added another sentence in the Brain MRI findings section to state that brain MRI anomalies did not differ in subgroups with deletion of the 15q11-q13 region and UPD (see pages 6-7).